# Identification of protein signatures for lung cancer subtypes based on BPSO method

**Jihan Wang**[1], **Hanping Wang**[1,2], **Jing Xu**[1], **Qiying Song**[1], **Baozhen Zhou**[1], **Jingbo Shangguan**[1], **Mengju Xue**[1]*, **Yangyang Wang**[3]*

1 Department of Basic Medicine, School of Medicine, Xi'an International University, Xi'an, 710077, China, 2 Engineering Research Center of Personalized Anti-aging Health Product Development and Transformation, Universities of Shaanxi Province, Xi'an, 710077, China, 3 School of Electronics and Information, Northwestern Polytechnical University, Xi'an, 710129, China

☯ These authors contributed equally to this work.
* xuemengju@163.com (MX); wangyang2154@mail.nwpu.edu.cn (YW)

## Abstract

The objective of this study was to identify protein biomarkers that can distinguish between LUAD and LUSC, critical for personalized treatment plans. The proteomic profiling data of LUAD and LUSC samples from TCPA database, along with phenotype and survival information from TCGA database were downloaded and preprocessed for analysis. We used BPSO feature selection method and identified 10 candidate protein biomarkers that have better classifying performance, as analyzed by t-SNE and PCA algorithms. To explore the causalities among these proteins and their associations with tumor subtypes, we conducted the PCStable algorithm to construct a regulatory network. Results indicated that 4 proteins, MIG6, CD26, NF2, and INPP4B, were directly linked to the lung cancer subtypes and may be useful in guiding therapeutic decision-making. Besides, spearman correlation, Cox proportional hazard model and Kaplan-Meier curve was employed to validate the biological significance of the candidate proteins. In summary, our study highlights the importance of protein biomarkers in the classification of lung cancer subtypes and the potential of computational methods for identifying key biomarkers and understanding their underlying biological mechanisms.

## Introduction

Lung cancer, as the most prevalent malignant disease worldwide, holds the top rank among cancers and has afflicted 2.2 million individuals, with 1.8 million fatalities [1]. The World Health Organization's (WHO) 2020 research reveals that the five-year survival rate in most countries is only 10% to 20%. Non-small-cell lung cancer (NSCLC), the primary form of lung cancer, is divided into two major subtypes, namely lung squamous cell carcinoma (LUSC) and lung adenocarcinoma (LUAD), together accounting for 90% of cases [2]. The consensus among experts is that LUAD and LUSC are more than just tumors with different histologic types. They have different biological signatures and clinical implications as well [3,4]. Previous studies have highlighted that these tumors may emerge from different epithelial cells, manifest unique cell markers, and display differing genomic profiles [3]. Distinct behaviors are

**Data Availability Statement:** The raw data and source code of BPSO we used in this research were uploaded to GitHub, which can be found in https://github.com/jihanwang/BPSO/.

**Funding:** This work was funded by Science and Technology Planning Project of Xi 'an (No. 22NYYF054); Engineering Research Center of Personalized Anti-aging Health Product Development and Transformation, Universities of Shaanxi Province (No. GCZX202201); the Key Research and Development Program of Shaanxi Province (No. 2020NY-197); Science and Research Special Project of Education Department of Shaanxi Provincial Project (No. 22JK0527).

**Competing interests:** The authors have declared that no competing interests exist.

exhibited by different histopathological and biological features of lung cancers, and correspondingly, specific therapeutic approaches should be provided. Tumorigenic alterations have been linked to the modification in protein structure or the expression of particular transcription factors in numerous studies [5–7]. However, the precise biomarkers to be implemented in clinical settings remain ambiguous [8]. Thus, accurate classification of lung cancers into clinically significant subtypes is of utmost importance in making therapeutic decisions and demands immediate attention [9,10].

In recent decades, numerous comprehensive investigations have been carried out to explore the complexity of cancer phenotypes and genotypes, which have facilitated the comprehension of cancer development and decreased the likelihood of overdiagnosis. Proteomics has emerged as a critical tool in studying biological changes in cancer [11]. Immunohistochemistry (IHC) has been widely employed to detect specific protein markers that differentiate LUAD from LUSC [12]. For instance, TTF1/NKX2-1 and KRT7 are commonly used markers for LUAD, while cytokeratin 5, cytokeratin 6, p63, and SOX2 are associated with LUSC. However, these markers have limitations in terms of specificity and sensitivity, highlighting the need for further investigation and the discovery of additional biomarkers [13]. As it is a challenging field to classify cancer subtypes using a limited number of molecules, various methods based on different medical databases have been proposed for significant biomarker selection [14]. The Cancer Proteome Atlas (TCPA), which encompasses 32 cancer types and over 8000 cancer samples, is a powerful resource in comprehending the pathophysiology and therapy of various cancers using functional proteomics [15]. Recent research indicates that transcriptome and proteome analysis can lay a foundation for comprehensively understanding the molecular mechanisms of lung cancer and developing new therapeutic approaches [16].

Feature selection methods have become a critical technology for reducing high-dimensional datasets by obtaining an optimal feature subset through the removal of irrelevant information and have been widely adopted in cancer diagnosis. Typically, feature selection methods are referred to as filter, wrapper, and embedding. In order to assess the performance of selected features, the wrapper technique employs a classifier and a search mechanism. In a recent study [17], a novel hybrid feature selection and ensemble learning framework was used to overcome the high dimension and redundancy in multi-omics datasets. Utilizing translational models, they were able to identify major actionable proteins linked to clinical outcomes including survival time [18]. Employing reverse phase protein array (RPPA) data, a hybrid feature selection model distinguished between different stages of head and neck squamous cell carcinoma (HNSCC) [19]. a computational framework was proposed by Zhang et al. based on RPPA data to categorize patient samples into ten major cancer types with most of the proteins presenting cancer hallmarks [20]. The implementation of feature selection techniques has demonstrated that gaining concise and informative molecular features can be achieved more effectively than through typical statistical methods [10,21]. This is particularly important in the realm of cancer research, where combining feature selection with proteomics can guide clinicians in identifying the key biochemical signatures unique to different types of cancer.

This study comprises a comprehensive analysis of the protein expression dataset for lung cancer, which was obtained from the TCPA database, in addition to phenotype and survival data from The Cancer Genome Atlas (TCGA) database. Firstly, we presented an overview of the functional proteomic heterogeneity of LUAD and LUSC tumor samples. Secondly, we employed the wrapper feature selection method to identify candidate protein biomarkers that outperformed the original features in classifying the two major lung cancer subtypes. Moreover, we used Bayesian Networks (BN) to identify the direct impact factors that distinguish the two subtypes. Finally, we thoroughly evaluated the potential clinical implications of the candidate protein biomarkers for tumor progression and prognosis.

This study may narrow the gap in our current understanding of lung cancer by seeking to identify novel biomarkers that can improve the classification of LUAD and LUSC. The investigation of molecular differences and interplay between the two histotypes can contribute to a deeper understanding of lung cancer biology.

## Materials and methods

### Data acquisition and preprocessing

The "TCGA-PANCAN32-L4.zip" dataset is a comprehensive resource containing 32 cancer types, 258 proteins markers, and over 7000 tumor cases, which can be obtained from the TCPA portal (https://tcpaportal.org/tcpa/) [22] following the TCPA guidelines. Initially, we extracted the lung cancer subset from the original dataset and obtained 687 tumor cases, consisting of 362 samples for LUAD and 325 samples for LUSC. Next, we removed the protein markers that contained missing values ("NA") in more than half of the samples and filled in the "NA" values for six protein markers with their average values. Finally, we obtained a proteome profiling dataset comprising 687 tumor samples and 217 protein markers.

We downloaded the survival and clinical phenotype information for LUAD and LUSC tumor samples from the TCGA portal (https://tcga-data.nci.nih.gov/tcga/) [23] and merged this information with the protein expression data using their respective sample IDs for further analysis. The clinical characteristics of the LUAD and LUSC samples were summarized in Table 1.

**Table 1. Clinical characteristics of lung tumor samples.**

| Clinical characteristics | | Number of cases | |
|---|---|---|---|
| | | **LUAD** | **LUSC** |
| **Age at initial pathologic diagnosis (year)** | <65 | 165 | 111 |
| | ≧65 | 182 | 206 |
| | Not reported | 15 | 8 |
| **Gender demographic** | Male | 167 | 246 |
| | Female | 195 | 79 |
| **Race demographic** | Asian | 5 | 6 |
| | Black | 38 | 19 |
| | White | 273 | 239 |
| | Not reported | 46 | 61 |
| **Tumor stage** | Stage I | 189 | 153 |
| | Stage II | 87 | 110 |
| | Stage III | 65 | 57 |
| | Stage IV | 17 | 3 |
| | Not reported | 4 | 2 |
| **OS_status** | Alive | 210 | 187 |
| | Dead | 141 | 133 |
| | Not reported | 11 | 5 |
| **OS_time (day)** | Alive | 989.60±941.19 | 1134.82±996.06 |
| | Dead | 846.79±746.02 | 885.68±939.14 |
| **Total number of cases** | | 362 | 325 |

OS: Overall survival.

All the procedures of data preprocessing were performed based on "pandas" package [24] of Python 3.9.16.

## Proteome profiling analysis

Both t-stochastic neighbor embedding (t-SNE) [25] and Principal component analysis (PCA) are dimensionality reduction techniques commonly used in data visualization for exploring the structure of high-dimensional datasets [26]. PCA is a linear technique that transforms the data into a lower-dimensional space by maximizing the variance of the data points along each principal component. t-SNE, on the other hand, is a nonlinear technique that focuses on preserving the local structure of the data points while reducing the dimensionality [27].

In this study, t-SNE and PCA were used to visualize the distributions of LUAD and LUSC samples based on their proteome profiling data. The "FactoMineR" package [28] in R was used for PCA, while the "Rtsne" package [27] was used for t-SNE. By using these techniques, the high-dimensional proteome data were reduced to two or three dimensions, which allowed for the visualization of the overall distribution of the data and identification of any clustering patterns or outliers. These visualizations can provide insight into the similarities and differences between the samples, which can be useful for further analysis and interpretation of the data [10].

## Using binary particle swarm optimization identifying key protein signatures

Feature selection (FS) technologies are aim to reduce the complexity of learning algorithms for high-dimensional datasets and increase the readability of data [29]. By mimicking the movement and behavior of birds within a flock, the particle swarm optimization (PSO) algorithm has been developed as a highly efficient optimization technique [30]. It is a type of swarm intelligence algorithm that has been applied to feature selection (FS) problems in recent years. PSO is a global optimization algorithm that aims to find the optimal solution by iteratively adjusting a set of candidate solutions called particles.

The algorithm starts by randomly initializing a swarm of particles, each representing a possible feature subset. Each particle has a fitness value that indicates how well it performs on a given classification task [30]. The particles are then updated in each iteration based on their current position and velocity, as well as their previous best position and the global best position found by any particle in the swarm [31]. The classical PSO algorithm can be represented by the following formulas:

$$v_{i,j}(t+1) = wv_{i,j}(t) + c_1R_1(p_{best,i,j} - x_{i,j}(t)) + c_2R_2(g_{best,i,j} - x_{i,j}(t))$$
$$x_{i,j}(t+1) = x_{i,j}(t) + v_{i,j}(t+1)$$
(1)

where $i$ is the index of particles in the swarm ($i = 1,2,..,n$), and $j$ is the index of position in the particle ($j = 1,2,..,m$). $x_{i,j}$ is the position of the $i$th particle, and $v_{i,j}$ is the velocity of the $i$th particle. $c_1$ and $c_2$ are the acceleration weights, $w$ is the positive inertia weight. $R_1$ and $R_2$ are random values in [0,1].

The binary particle swarm optimization (BPSO) is modified from the original PSO algorithm, which introduces the concept of velocity as a probability that a bit takes on one or zero [32]. The rule for updating the position of BPSO are as follows:

$$x_{i,j}(t+1) = \begin{cases} 0, & \text{if } \text{rand}() \geq S(v_{i,j}(t+1)) \\ 1, & \text{if } \text{rand}() < S(v_{i,j}(t+1)) \end{cases}$$
(2)

$$S(v_{i,j}(t+1)) = \frac{1}{1 + e^{-v_{i,j}(t+1)}}$$
(3)

In the above formula, the sigmoid function of S for transforming the velocity to the probability.

BPSO computes faster than PSO with a better robustness and is suitable for feature selection task of providing an excellent solution in medical or biological studies. The code of BPSO for this study was modified based on https://github.com/NajiAboo/BPSO_BreastCancer. We also employed signal-to-noise ratio (SNR) filter feature selection method [33] for data preprocessing to speed up the calculation of BPSO. The equation of SNR was as follow:

$$SNR = \frac{\mu_1 - \mu_2}{\sigma_1 + \sigma_2} \tag{4}$$

where $\mu_1$ and $\mu_2$ are the means of LUAD and LUSC respectively. $\sigma_1$ and $\sigma_2$ refers to the standard deviation from the two groups.

## Bayesian networks

As a kind of probabilistic graphical model, Bayesian Networks (BN) has become popular in knowledge discovery and causal reasoning [34]. The structure $G$ of BN is a directed acyclic graph that expresses conditional independencies and dependencies among random variables associated with each node. The join probability distribution over a vector of random variables $X = (X_1, ..., X_n)$ of a BN can be factorized as follows:

$$P(X_1, ..., X_n) = \prod_{i=1}^{n} P(X_i | Pa(X_i^G)) \tag{5}$$

where $Pa(X_i^G)$ is the parent nodes of $X_i$.

In this study, we performed BN structure learning algorithm of PCStable [35] to build a regulatory network for exploring the direct protein biomarker associated with the classification of LUAD and LUSC. The PCStable source code of MATLAB can be found in https://github.com/z-dragonl/Causal-Learner, and the visualization of BN was achieved using Cytoscape 3.9.1 [36].

## Statistical analysis

The wilcoxon rank sum test [37] is an analytical tool that compares two independent samples while the Spearman correlation [38] assesses the strength and direction of the monotonic association between variables. The comparison of protein abundances between two cancer types of LUAD and LUSC were preformed using wilcoxon rank sum test, with p-value < 0.05 indicates statistically significant. We conducted spearman correlation analysis for evaluating the strength and direction of correlation between protein abundance and tumor Stage (Stage I, II, III, IV) and the correlation is regarded as significant when p-value < 0.05. In determining the factors critical for the prognosis of the tumor, we carried out Univariate Cox regression analysis, employing the Cox Proportional-Hazards Model (with coxph function in the "survival" package [39]) based on R. We generated Kaplan-Meier survival curve with "survminer" package [40] for estimating the survival function from lifetime data.

## Results

### The distributions overview of the LUAD and LUSC tumor samples

To gain insights into the distribution and heterogeneity of tumor samples with respect to two subtypes of lung cancers, we utilized t-SNE and PCA to visualize the profiling of a dataset

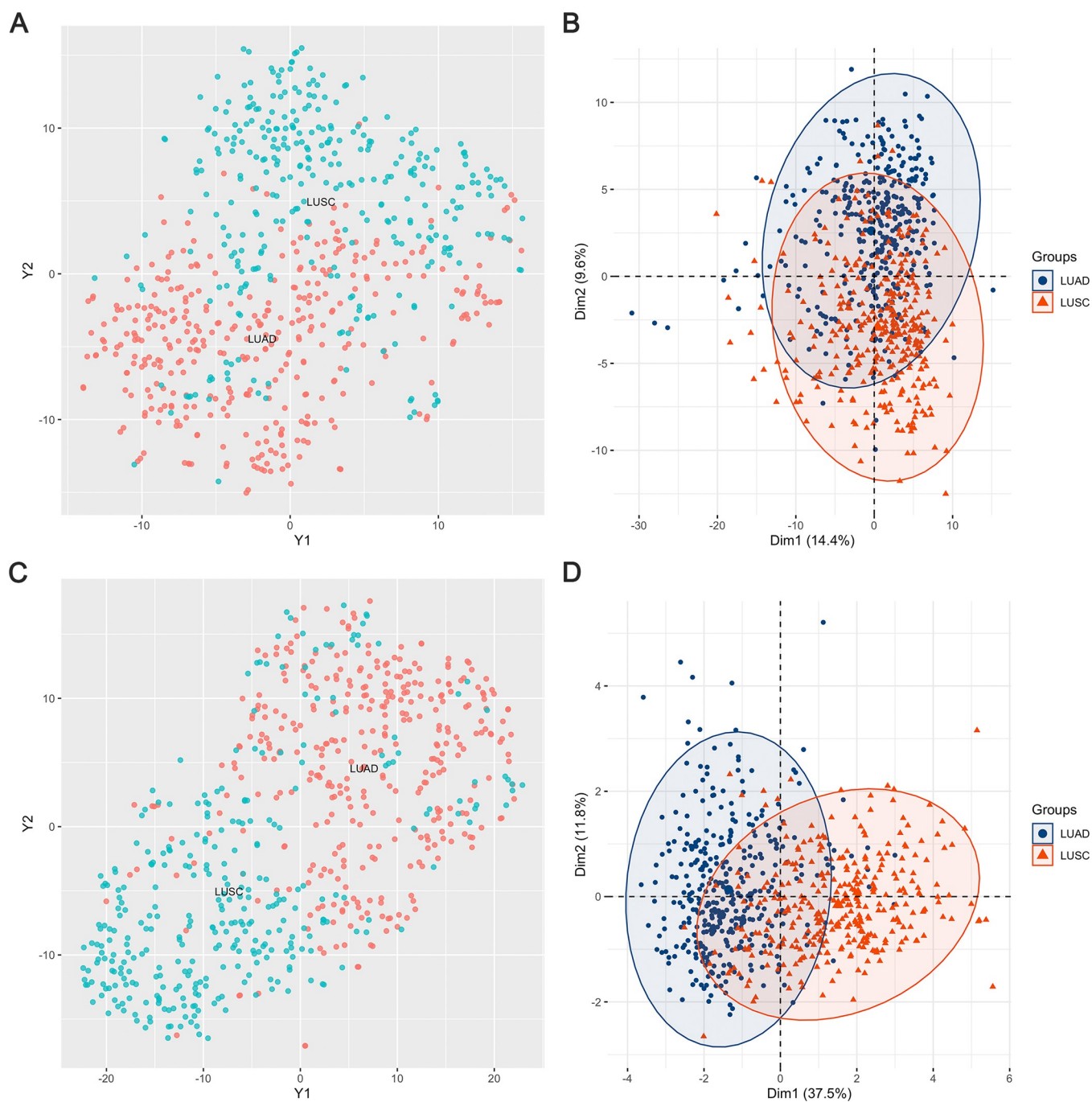

**Fig 1.** Overview of the proteome profiling across LUAD and LUSC tumor samples: Clusters of samples based on 217 proteins profiling were observed through t-SNE and PCA in (**A**) and (**B**), whereas for the 10 selected protein profiling, clusters were shown through t-SNE and PCA in (**C**) and (**D**).

containing 687 tumor samples and 217 protein markers. As shown in Fig 1A and 1B, the clusters of two subtypes of tumor samples were not distinguished clearly. Specifically, the PCA scores for PC1 (Dim1) was 14.4% and for PC2 (Dim2) was 9.6%, which indicated the overlap of LUAD and LUSC was significant with each other.

## Using BPSO method identifying protein signatures for classifying the two cancer subtypes

For obtaining the significant biomarkers of tumor samples classification, we applied the SNR and BPSO implemented by Python to the candidate dataset based on MacOS with a 1.6-GHz CPU and 8 GB of memory. The number of particle population and iteration were set to 500 and 50, respectively. Meanwhile, the acceleration weight of $c_1$ was set to 1 and 2 for $c_2$, the inertia weight $w$ was set to 0.5. Based on the principle of SNR > 0.4, we removed 198 proteins from 217 original protein dataset and 19 proteins were remained for further analysis. After performing BPSO method, 10 candidate proteins including TFRC, CD26, MIG6, GAPDH, INPP4B, FOXM1, NF2, ACVRL1, IGFBP2, and X4EBP1 were extracted from the previous 19 proteins.

In addition, we also performed differential protein analysis based on the edgeR package. The results showed that four out of the 10 most significantly differential proteins were also belonged to the 10 protein markers obtained by the BPSO method, including GAPDH, TFRC, IGFBP2 and INPP4B. The detail differential protein analysis result was shown in S1 Table, which contained the 10 most significantly differential proteins between LUAD vs. LUSC.

Fig 2 revealed that according to the Wilcoxon rank sum test, there were considerable differences in protein abundances between LUAD and LUSC tumor samples. Detailed information about the 10 selected protein abundances was summarized in S2 Table. We employed t-SNE and PCA analysis based on the 10 selected protein markers, and obtained better clustering effect, as shown in Fig 1C and 1D. Compared with Fig 1B for PCA, the value of Dim1 increased from 14.4% to 37.5%. Thus, BPSO was able to improve identifying markers/signatures in a high-dimensional data.

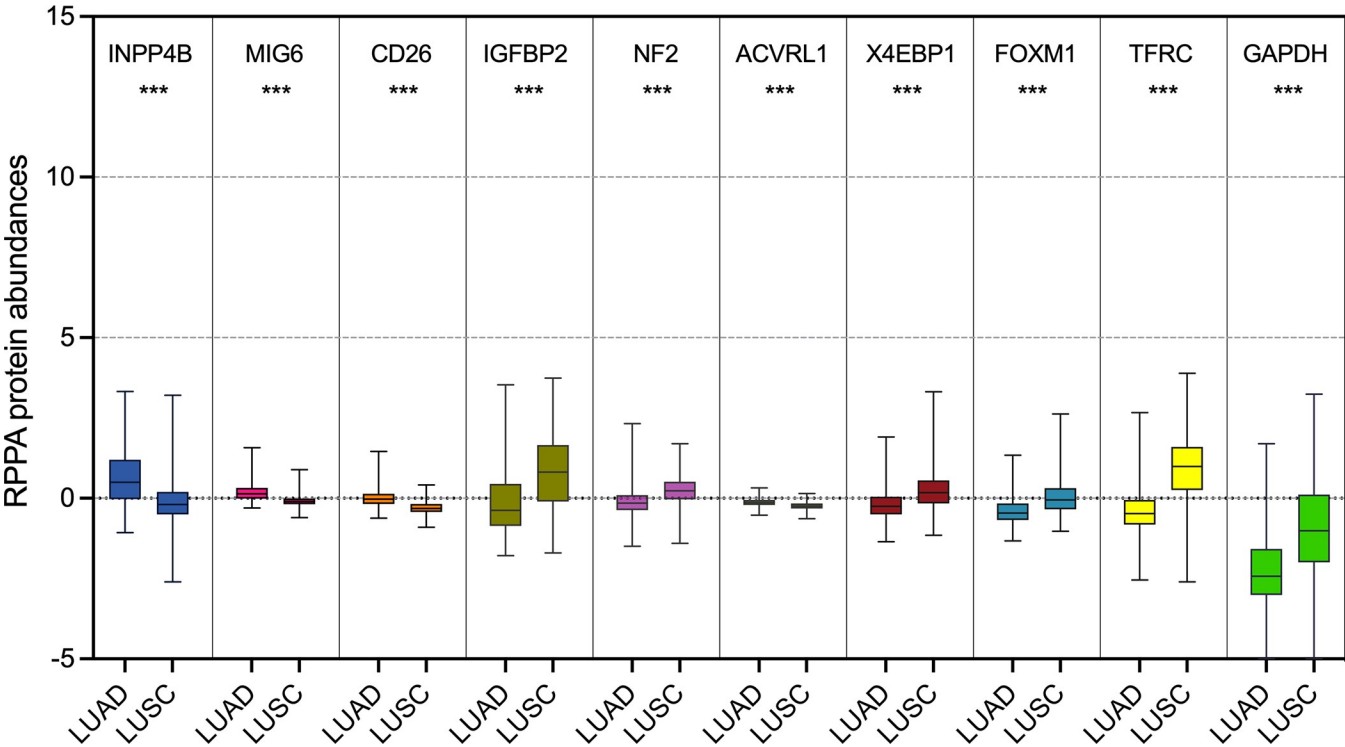

**Fig 2. The relative abundances of the selected 10 protein signatures for LUAD and LUSC tumor samples.** The minimum, median, and maximum protein abundance for each tumor type is illustrated in each box plot. Wilcoxon Rank Sum Test indicated *** $P < 0.0001$.

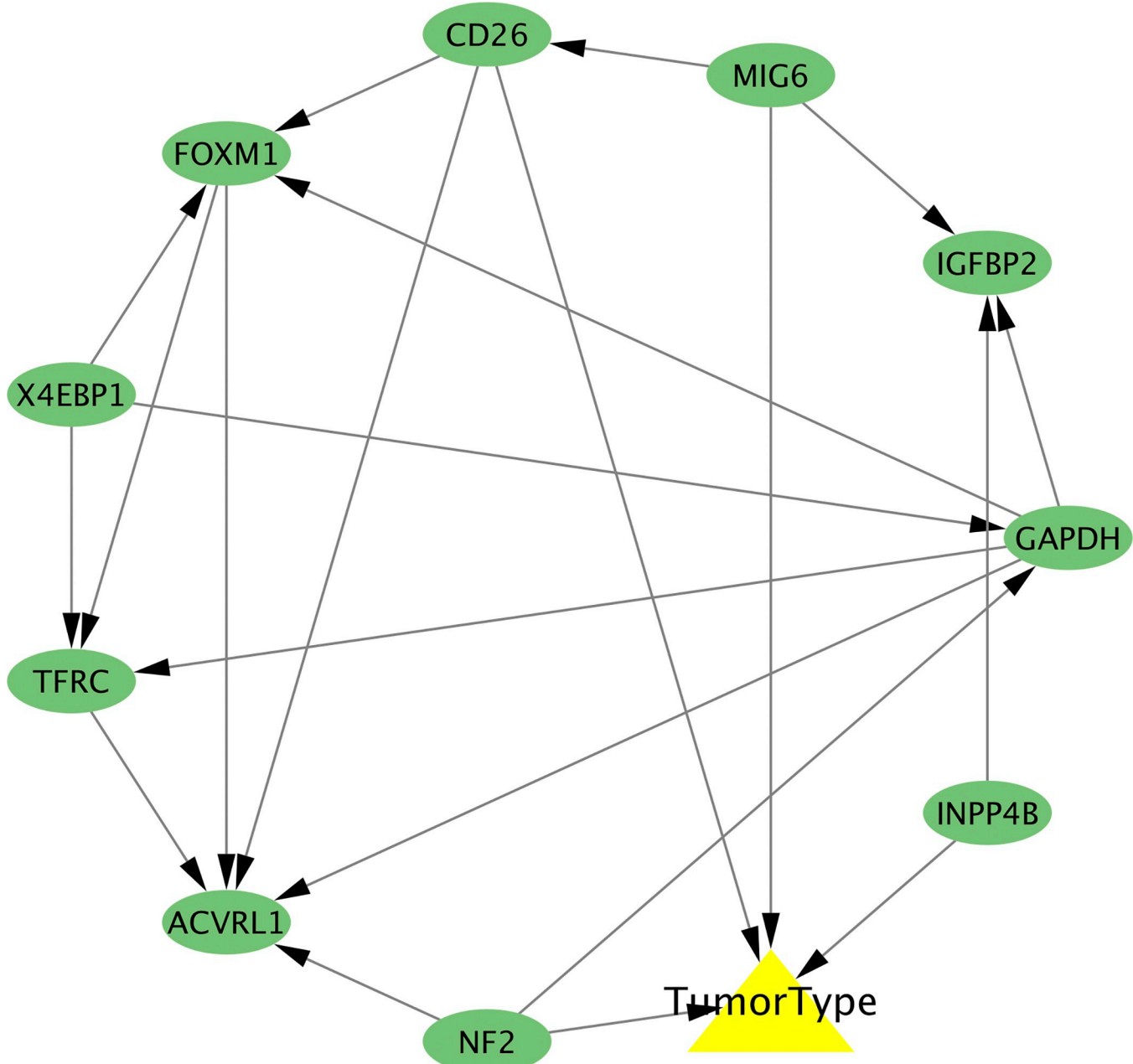

**Fig 3. The Bayesian networks generated by PCStable algorithm for 10 candidate protein signatures.**

### Using Bayesian networks to discover the causalities

Bayesian networks have a diverse range of application such as medical and biological research. As shown in Fig 3, each of the 10 nodes with green color corresponded to the 10 protein bio-markers, and the triangle node with yellow triangle was the tumor type. The links with arrows represented that one node directly influenced the other node. We observed that the proteins of MIG6, CD26, NF2 and INPP4B were the direct impact factors for the lung cancer subtypes classification. MIG6 is critical for lung cancer due to its controlling the regulation of cell

signaling [41]. The study revealed that CD26 serves as a valuable prognostic marker and a promising therapeutic target for lung cancer [42].

The four candidate protein signatures also were used for t-SNE and PCA (S1 Fig). The profiling of t-SNE based on the four signatures was comparable to the profiling based on the 10 signatures, and the result of PCA showed better classification performance with the score of 65.6% (Dim1 was 44.3% and Dim2 was 21.3%).

**The correlation between protein biomarkers and clinical characteristics of tumors.** To evaluate the potential significance of these proteins in tumor progression or prognosis, we analyzed the associations between selected protein biomarkers and tumor stage as well as overall survival (OS) status. Fig 4A and 4B illustrated the positive correlation between TFRC and FOXM1 expressions and tumor stages, as well as their association with tumor stages in LUAD samples using spearman correlation ($P < 0.05$). Similarly, LUSC samples showed a positive correlation between MIG6 and tumor stage, as depicted in Fig 4C and 4D using spearman correlation.

We performed Cox proportional hazard model to identify risk clinical parameters of tumor patients. As shown in Fig 5, results from univariate Cox regression analysis revealed that

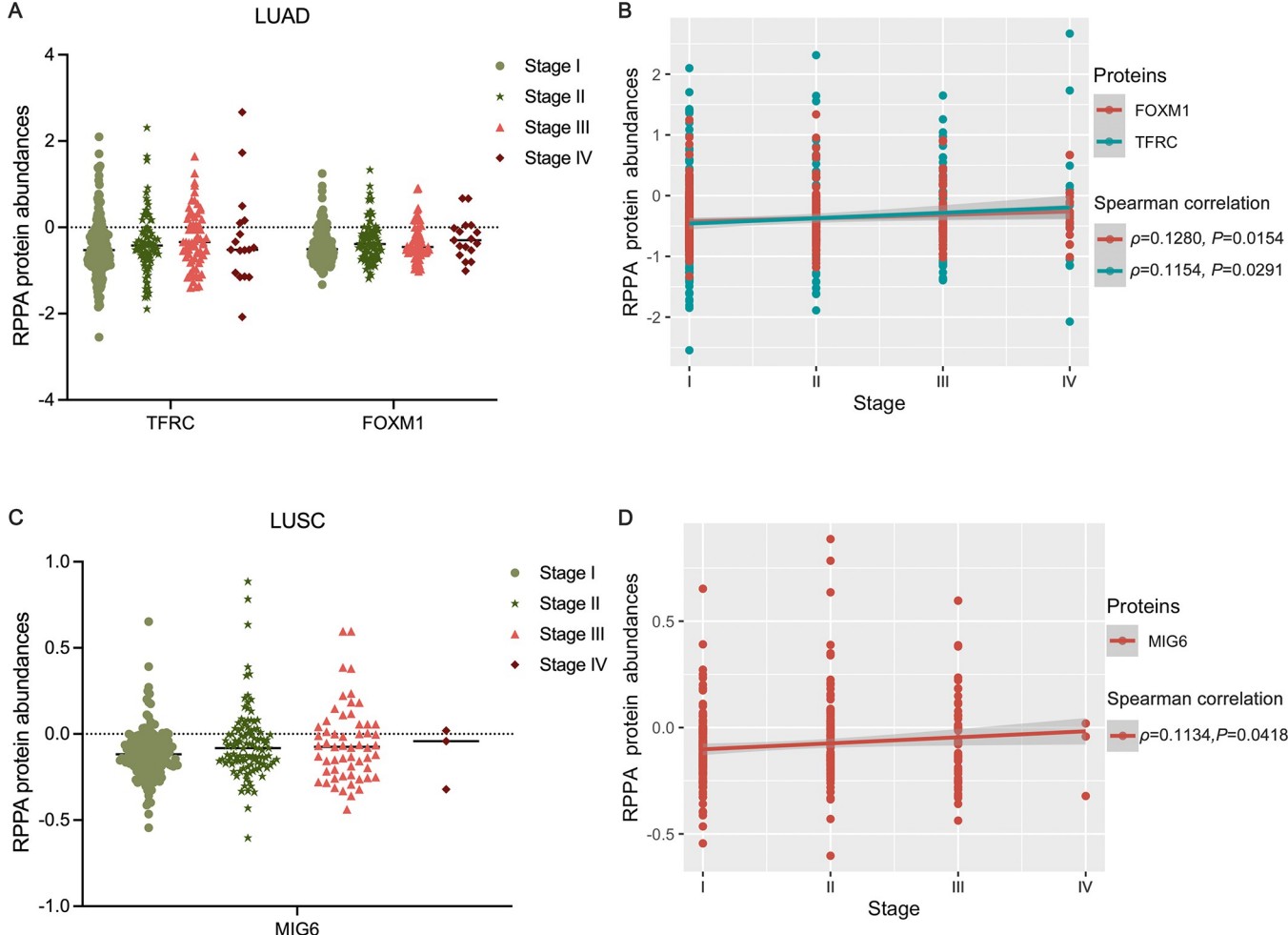

**Fig 4. The relationship between protein biomarkers and tumor stage.** (**A**) scatter plot of proteins TFRC and FOXM1 in stages I/II/III/IV of LUAD tumors; (**B**) spearman correlation of proteins TFRC and FOXM1 with tumor stage in LUAD; (**C**) scatter plot of proteins MIG6 in stages I/II/III/IV of LUSC tumors; (**D**) spearman correlation of proteins MIG6 with tumor stage in LUSC.

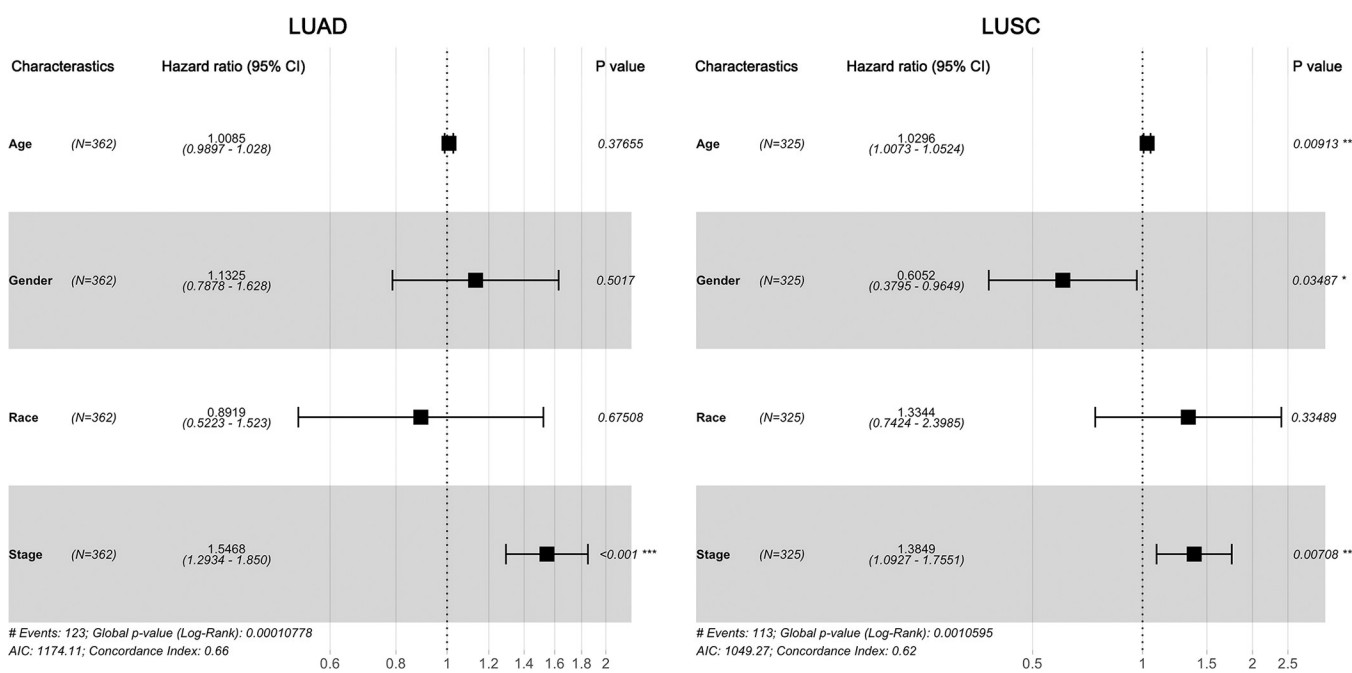

**Fig 5. Forest plots of univariate Cox regression analysis in LUAD and LUSC.**

tumor stage was a significant risk factor for poor overall survival in both LUAD (HR = 1.5468, $P < 0.05$) and LUSC (HR = 1.3849, $P < 0.05$) patients. Furthermore, higher age was linked to higher prognostic risk in LUSC patients (HR = 1.0296, $P < 0.05$), and the gender factor affect the overall survival that males had a lower survival rate than female in LUSC (HR = 0.6052, $P < 0.05$).

Kaplan-Meier (KM) curves was used to estimate the potential factors that affect the fraction of patients living for a certain amount of time on the basis of candidate proteins. As shown in Fig 6A, the protein biomarker of FOXM1 was positively correlated with the LUAD prognosis ($P < 0.05$), and GAPDH was positively correlated with the prognosis for LUSC ($P < 0.05$, Fig 6B).

## Discussion

Lung carcinoma is a primary cause of cancer-related mortality worldwide. Proper identification and classification of the various subtypes of lung carcinoma are crucial to enhance diagnosis, treatment, and patient outcomes. In the domain of lung carcinoma, protein biomarkers have gained increasing importance for the classification of subtypes based on the expression of specific proteins in tumor cells [43,44]. As an indispensable component of numerous biological data analysis pipelines, feature selection methods present a powerful tool for identifying the most informative protein biomarkers [45].

In this study, we characterized the profiles of 687 lung tumor samples with 217 proteins, and we observed that the two subtypes of tumor samples could not be distinguished based on t-SNE and PCA. This indicated that using all the original protein features for classification may not be advisable, particularly for high-dimensional biological data. Hence, it is necessary to explore key protein biomarkers that can identify the two subtypes of lung carcinoma. BPSO presents a fast convergence and low computational cost feature selection method that can handle discrete and binary optimization problems [46,47]. By utilizing the BPSO algorithm, we

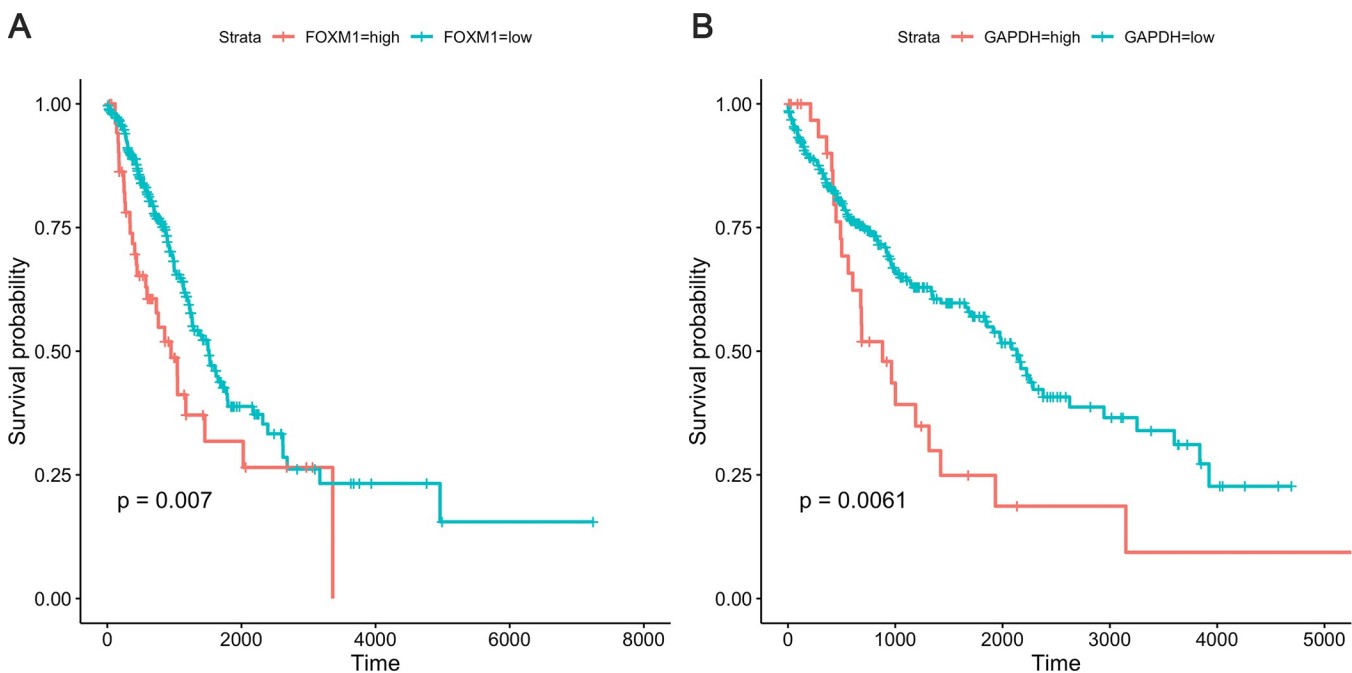

**Fig 6. Kaplan-Meier survival curves based on the candidate proteins: The curves indicate that (A) FOXM1 and (B) GAPDH have a significant impact on the overall survival rate of patients with LUAD and LUSC, respectively.**

obtained 10 candidate protein biomarkers for better distinguishing the two subtypes. The differential protein analysis also performed and there were 4 proteins with significant difference that overlapped with the 10 candidate biomarkers. We used Wilcoxon rank-sum test to verify the significant differences in protein abundances between LUAD and LUSC tumor samples based on the 10 selected biomarkers. The classification performance of the 10 protein signatures outperformed that of all proteins, as evidenced by t-SNE and PCA. While feature selection can provide a few biomarkers that improve classification, it cannot explore the causalities essential for knowledge discovery. Therefore, it is necessary to build models, such as Bayesian networks, to explore more causalities. The PCStable algorithm [48] is a powerful and flexible method for learning the structure of Bayesian networks from data. Its ability to handle both discrete and continuous variables, and to learn stable network structures, makes it particularly useful in many real-world applications. Based on the 10 candidate proteins, we utilized PCStable to construct the regulatory network for exploring the associations among these proteins and the tumor subtype classification. Understanding the molecular biomarkers associated with specific tumor subtypes can aid in personalized treatment planning. We noticed that the 4 proteins of MIG6, CD26, NF2, and INPP4B linked the lung carcinoma subtypes directly, indicating that these 4 protein biomarkers may aid in distinguishing between different subtypes of tumors and guide therapeutic decision-making. The 4 candidate proteins were also employed for classifying the subtypes of lung cancer, and the results showed better classification performance compared to the profiling of 10 candidate biomarkers.

These studies using feature selection algorithms to filter protein biomarkers for cancer subtype classification based on the TCPA database are rare. We compared the results of this paper with similar studies. Wang et.al performed three feature selection methods (pyHSICLasso, XGBoost, and Random Forest) to select protein features for classifying cancer subtypes [49]. For lung tumor subtype classification, 20 candidate protein biomarkers were obtained, among

which 8 protein biomarkers, TFRC, CD26, MIG6, GAPDH, INPP4B, FOXM1, NF2, and IGFBP2, were also captured by the method of this study. The study [50] aimed to search for prognostic biomarkers of stomach adenocarcinoma (STAD) by using a prognostic risk model based on Cox regression analysis, and tri-protein was identified as an independent prognostic factor. The idea of identifying prognostic risk clinical parameters by COX analysis is consistent with our study.

Furthermore, our results were partially consistent with prior research. Numerous studies have established the usefulness of MIG6 as a biomarker and therapeutic target for lung carcinoma, and the overexpression of MIG6 can restrict the oncogenic transformation of non-small-cell lung carcinoma (NSCLC) cells [41,51]. FoxM1 has also been shown to be significant in various tumor initiations and progressions. High expression of FoxM1 in NSCLC patients is correlated with poor prognosis [52]. CD26/DPP4 is a transmembrane glycoprotein, and studies have shown that lung cancer patients exhibit four times greater CD26/DPP4 activity than normal tissue. The inhibition of CD26/DPP4 can suppress lung cancer growth in mice [53]. In summary, our emphasis is on using feature selection in high-throughput biological data analysis, especially in cancer biomarker research, which is beneficial in developing personalized treatment strategies with clinical significance.

In our study, a comprehensive proteomic analysis was employed, utilizing a large-scale dataset of lung cancer samples and powerful feature selection method as well as Bayesian Network. This integration facilitated the identification of candidate protein biomarkers. The limitations of our study are those characteristics of limited clinical data and the lack of biomarker functional validation. The protein datasets and clinical information used in this study were limited and contained missing values, which had an impact on data preprocessing and feature selection methods and limited our ability to conduct more detailed analyzes between identified biomarkers and clinical outcomes. Correlation analysis capabilities. More convincing results require more complete lung tumor samples to support. Additionally, Beyesian network has its inherent limitations that the intrinsic causal relationships based on dataset with small samples may not be figured out, and the candidate biomarkers roles in lung cancer progression and response to treatment remain to be fully evaluated based on further experimental studies.

## Conclusions

The utilization of biomarkers for the classification of lung cancer subtypes is a crucial field of study that holds significant potential for enhancing the diagnosis, treatment, and outcomes for patients afflicted with lung cancer. Feature selection approach and comprehensive large dataset employed in our study have allowed for the identification of key biomarkers associated with different subtypes of lung cancer. A robust and reliable framework for the identification and validation of these biomarkers has shown by using these computational algorithms and statistical analyses. These identified biomarkers can serve as reliable diagnostic markers to distinguish between LUAD and LUSC, aiding in accurate classification and subsequent treatment decisions. Furthermore, the findings of this study have implications for personalized medicine and targeted therapies. By identifying biomarkers specific to each subtype, it becomes possible to develop targeted therapeutic strategies tailored to the molecular characteristics of individual patients.

## Supporting information

**S1 Fig. Overview of the proteome profiling across LUAD and LUSC tumor samples by using MIG6, CD26, NF2, and INPP4B. (A)** t-SNE showed the clusters of samples based on 4

proteins profiling. (**B**) PCA showed the clusters of samples based on 4 proteins profiling.
(TIF)

**S1 Table. Differential protein analysis between LUAD vs. LUSC.**
(DOCX)

**S2 Table. Analysis of the 10 selected proteins between LUAD vs. LUSC.**
(DOCX)

## Author Contributions

**Conceptualization:** Jihan Wang, Mengju Xue, Yangyang Wang.

**Data curation:** Jing Xu, Jingbo Shangguan.

**Formal analysis:** Yangyang Wang.

**Funding acquisition:** Hanping Wang.

**Investigation:** Qiying Song.

**Methodology:** Jihan Wang, Jing Xu, Yangyang Wang.

**Validation:** Hanping Wang, Jing Xu, Qiying Song, Baozhen Zhou, Jingbo Shangguan.

**Visualization:** Yangyang Wang.

**Writing – original draft:** Jihan Wang.

**Writing – review & editing:** Baozhen Zhou, Mengju Xue, Yangyang Wang.

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
