## [Decision Letter · Decision Letter 0]

27 Sep 2023

PONE-D-23-21264Identification of protein signatures for lung cancer subtypes based on BPSO methodPLOS ONE

Dear Dr. Wang,

Thank you for submitting your manuscript to PLOS ONE. After careful consideration, we feel that it has merit but does not fully meet PLOS ONE’s publication criteria as it currently stands. Therefore, we invite you to submit a revised version of the manuscript that addresses the points raised during the review process.

We look forward to receiving your revised manuscript.

Kind regards,

Bingli Wu

Academic Editor

PLOS ONE

Journal Requirements:

2. Please note that PLOS ONE has specific guidelines on code sharing for submissions in which author-generated code underpins the findings in the manuscript. In these cases, all author-generated code must be made available without restrictions upon publication of the work. 

Please review our guidelines at https://journals.plos.org/plosone/s/materials-and-software-sharing#loc-sharing-code and ensure that your code is shared in a way that follows best practice and facilitates reproducibility and reuse.

https://www.mdpi.com/2218-273X/13/4/701/htm

In your revision ensure you cite all your sources (including your own works), and quote or rephrase any duplicated text outside the methods section. Further consideration is dependent on these concerns being addressed.

5.  Thank you for stating the following financial disclosure: "Yes"

6. Thank you for stating the following in your Competing Interests section: "NO authors have competing interests"

7. In your Data Availability statement, you have not specified where the minimal data set underlying the results described in your manuscript can be found. PLOS defines a study's minimal data set as the underlying data used to reach the conclusions drawn in the manuscript and any additional data required to replicate the reported study findings in their entirety. All PLOS journals require that the minimal data set be made fully available. For more information about our data policy, please see http://journals.plos.org/plosone/s/data-availability.

8. PLOS requires an ORCID iD for the corresponding author in Editorial Manager on papers submitted after December 6th, 2016. Please ensure that you have an ORCID iD and that it is validated in Editorial Manager. To do this, go to ‘Update my Information’ (in the upper left-hand corner of the main menu), and click on the Fetch/Validate link next to the ORCID field. This will take you to the ORCID site and allow you to create a new iD or authenticate a pre-existing iD in Editorial Manager. Please see the following video for instructions on linking an ORCID iD to your Editorial Manager account: https://www.youtube.com/watch?v=_xcclfuvtxQ

Reviewers' comments:

Reviewer's Responses to Questions

**Comments to the Author**

1. Is the manuscript technically sound, and do the data support the conclusions?

Reviewer #1: Yes

Reviewer #2: Partly

2. Has the statistical analysis been performed appropriately and rigorously? 

Reviewer #1: Yes

Reviewer #2: Yes

3. Have the authors made all data underlying the findings in their manuscript fully available?

Reviewer #1: Yes

Reviewer #2: Yes

4. Is the manuscript presented in an intelligible fashion and written in standard English?

Reviewer #1: Yes

Reviewer #2: Yes

5. Review Comments to the Author

Reviewer #1: The aim of this article is to identify protein biomarkers that can distinguish LUAD from LUSC, which is essential for personalized treatment regimens. Using the BPSO feature selection method, the authors identified 10 candidate protein biomarkers with good classification performance by t-SNE and PCA algorithm analysis. To explore the causal relationship between these proteins and their association with tumor subtypes, a regulatory network was constructed using the PCStable algorithm. The authors also used Mauman's correlation, Cox proportional hazards models, and Kaplan-Meier curves to verify the biological significance of the candidate proteins. However, I have some suggestions and questions about certain parts of the manuscript.

1. According to what criteria are the authors selected 10 candidate protein features by BPSO algorithm?

2. Whether NF2 and INPP4B have been demonstrated as prognostic markers in lung cancer.

3. The authors have repeatedly described that the feature selection method BPSO is superior to other methods, but the speed, computational cost and robustness of other methods are not compared in the results.

4. The experimental verification of the four proteins as lung cancer biomarkers is insufficient, and it is recommended to add more experiments to verify the effects of the four proteins on lung cancer.

Reviewer #2: The overall idea of identifying protein markers in order to differentiate lung adenocarcinoma and squamous cell carcinoma is beneficial and important. The results were obtained using the protein expression dataset for lung cancer from the public TCPA database.

Major concerns are in the following points:

1. In the introduction and in the discussion, the authors claimed they “identify candidate protein biomarkers that outperformed the original features in classifying the two major lung cancer subtypes." and “The classification performance of the 10 protein signatures outperformed that of all proteins, as evidenced by t-SNE and PCA.”

a. Both t-SNE and PCA are graphical visualization approaches, but no quantitative measurements (e.g. sensitivity, specificity, AUC) were used to evaluate the classification accuracy for adenocarcinoma vs squamous cell carcinoma.

b. The authors didn’t compare the classification accuracy of their method to the prior outstanding approaches. For instance, the prior study (Ref. 8: DOI: 10.1038/modpathol.2011.92) demonstrated that “a two-marker panel of TTF-1/p63 is sufficient for subtyping of the majority of tumors as adenocarcinomas vs squamous cell carcinoma”. Did the authors compare the predictive performance of the selected 10 or 4 protein signatures with the two-marker panel of TTF-1/p63?

c. Moreover, the proposed BPSO algorithm is a feature selection approach, the classification accuracy by using the selected features should be validated by using cross-validation and/or on other independent datasets.

2. In Figure 3, the network analysis identified 4 proteins “directly linked” to the classification of adenocarcinoma vs squamous cell carcinoma. It doesn’t currently make scientific sense how the four proteins will directly impact the subtype classification. Please discuss this, otherwise it should be discussed as a limitation.

Minor:

Legends or contents in some figures are too small, or low resolution (e.g. Figures 1 and 5).

6. PLOS authors have the option to publish the peer review history of their article (what does this mean?). If published, this will include your full peer review and any attached files.

Reviewer #1: No

Reviewer #2: No

---

## [Author Response · Author response to Decision Letter 0]

9 Oct 2023

All responses were uploaded to the submission system.

---

## [Editor Report · Decision Letter 1]

27 Oct 2023

Identification of protein signatures for lung cancer subtypes based on BPSO method

PONE-D-23-21264R1

Dear Dr. Wang,

We’re pleased to inform you that your manuscript has been judged scientifically suitable for publication and will be formally accepted for publication once it meets all outstanding technical requirements.

Kind regards,

Bingli Wu

Academic Editor

PLOS ONE
---

## [Editor Report · Acceptance letter]

28 Nov 2023

PONE-D-23-21264R1 

Identification of protein signatures for lung cancer subtypes based on BPSO method 

Dear Dr. Wang:

I'm pleased to inform you that your manuscript has been deemed suitable for publication in PLOS ONE. Congratulations! Your manuscript is now with our production department. 

Kind regards, 

on behalf of

Dr. Bingli Wu 

Academic Editor

PLOS ONE